# Identification and DNA Marker Development for a Wheat-*Leymus mollis* 2Ns (2D) Disomic Chromosome Substitution

**DOI:** 10.3390/ijms23052676

**Published:** 2022-02-28

**Authors:** Xianbo Feng, Xin Du, Siwen Wang, Pingchuan Deng, Yongfu Wang, Lihui Shang, Zengrong Tian, Changyou Wang, Chunhuan Chen, Jixin Zhao, Wanquan Ji

**Affiliations:** 1State Key Laboratory of Crop Stress Biology for Arid Areas and College of Agronomy, Northwest A&F University, Yangling, Xianyang 712100, China; fxb1201@163.com (X.F.); duxin060031@nwafu.edu.cn (X.D.); wangsiwen2018@nwafu.edu.cn (S.W.); dengpingchuan@nwsuaf.edu.cn (P.D.); wyf063575@163.com (Y.W.); slh9707@163.com (L.S.); tian.zr@163.com (Z.T.); chywang2004@126.com (C.W.); chchch8898@163.com (C.C.); 2Shaanxi Research Station of Crop Gene Resources and Germplasm Enhancement, Ministry of Agriculture, Yangling, Xianyang 712100, China

**Keywords:** *Leymus mollis*, molecular cytogenetics, SLAF-seq, stripe rust, wheat 55K array

## Abstract

*Leymus mollis* (2*n* = 4*x* = 28, NsNsXmXm), a wild relative of common wheat (*Triticum aestivum* L.), carries numerous loci which could potentially be used in wheat improvement. In this study, line 17DM48 was isolated from the progeny of a wheat and *L. mollis* hybrid. This line has 42 chromosomes forming 21 bivalents at meiotic metaphase I. Genomic in situ hybridization (GISH) demonstrated the presence of a pair chromosomes from the Ns genome of *L. mollis*. This pair substituted for wheat chromosome 2D, as shown by fluorescence in situ hybridization (FISH), DNA marker analysis, and hybridization to wheat 55K SNP array. Therefore, 17DM48 is a wheat-*L. mollis* 2Ns (2D) disomic substitution line. It shows longer spike and a high level of stripe rust resistance. Using specific-locus amplified fragment sequencing (SLAF-seq), 13 DNA markers were developed to identify and trace chromosome 2Ns of *L. mollis* in wheat background. This line provides a potential bridge germplasm for genetic improvement of wheat stripe rust resistance.

## 1. Introduction

Common wheat (*Triticum aestivum L.*, 2*n* = 6*x* = 42, AABBDD) is one of the major food crops in the world. Against the background of the prominent contradiction between population growth and the shortage of resources, wheat yield is increasingly challenged. Wheat stripe rust, caused by the fungus *Puccinia*
*striiformis f.* sp. *tritici* (*Pst*), is a serious threat to wheat production. When wheat is susceptible to *Pst* infection, the yield loss is about 10–20%, but it can reach 50% or even result in no harvest in a pandemic year [1]. The most economical, effective and environmental-friendly method is to breed resistant cultivars, but the narrow genetic basis of wheat restricts its genetic improvement [2]. Therefore, it is necessary to provide new wheat germplasms with stripe rust resistance to facilitate wheat resistance breeding.

*Leymus mollis* (Trin.) Pilger (2*n* = 4*x* = 28, NsNsXmXm), a wild relative of wheat, possesses large and long spike with numerous spikelets and resistance to multiple abiotic and biotic stresses [3,4]. Through wild hybridization and cytogenetic manipulation, different wheat-*L. mollis* derivatives were developed as bridge materials for wheat improvement. Among them, octaploid *Tritileymus* line M842 (2*n* = 8*x* = 56, AABBDDNsNs) carrying many beneficial agronomic traits were identified through cytogenetic methods [5]. Yield and resistance-related genes have been transferred into wheat background in the form of partial amphiploidy, addition lines, substitution lines and translocation lines [6,7,8]. The accurate identification and rapid trace of alien chromosomes or translocated fragments directly affect breeding process [9]. Molecular markers are one of the most convenient methods to identify alien chromosomes due to the high efficiency. The existing markers cannot meet the needs of *L. mollis* chromosome detection in the common wheat background. With the development of next generation sequencing technology and bioinformatics tools, SLAF-seq (specific-locus amplified fragment sequencing) technique has been developed as a high-resolution strategy for large scale de novo discovery and genotyping of SNP [10]. It also provides a strategy for developing markers to detect and track alien chromosomes or genes from wheat wild relatives in the wheat background. Based on the SLAF-seq technology, 507 STS markers of *Thinoprum ponticum* 1Js chromosomes, 89 specific markers of *Thinopyrum elongatum* 7E chromosomes, two 2St-chromosome-specific markers, and two 3St-chromosome-specific markers were developed respectively [11,12,13]. Therefore, SLAF-seq is a feasible method to develop sufficient markers for accurate detection of alien chromatin, which could promote the further exploitation of beneficial genes for wheat genetic improvement. To date, no specific SLAF-based markers have been reported in *L. mollis*.

In the study, a wheat-*L. mollis* 2Ns (2D) disomic substitution line 17DM48 with a good level of stripe rust resistance and longer spike was identified from the derivatives of wheat and *L. mollis*. The chromosome numbers of 17DM48 and its chromosomal inherited behavior in meiosis were surveyed by cytogenetic methods. The chromosomal composition was ascertained using FISH, GISH, molecular markers and a single-nucleotide polymorphism (SNP) array. Specific markers were developed using SLAF-seq for tracing chromosome 2Ns and improving the selection efficiency. Its agronomic traits and stripe rust resistance were evaluated in the field experiment. This line could serve as a bridge material for wheat genetic improvement to increase stripe rust resistance.

## 2. Results

### 2.1. Cytological Characterization of 17DM48

Root tip cells (RTCs) in mitosis metaphase, pollen mother cells (PMCs) in meiosis metaphase I and anaphase I were observed and counted respectively using an Olympus BX-43 microscope. Of 150 RTCs, 143 (95.33%) cells contained 42 chromosomes, showing that 17DM48 had the chromosome numbers with 2*n* =42 (Figure 1a). A total of 82 PMCs were observed, 78 PMCs had the chromosome configuration of 2*n* = 21II (Figure 1b). No trivalents or quadrivalents were observed at metaphase I, and no chromosomes were lagged at anaphase I (Figure 1c). Consequently, line 17DM48 exhibited high cytological stability.

### 2.2. GISH and Sequential FISH–GISH Analysis

Genomic DNA of *L. mollis* and *P. huashanica* were used as probe respectively to identify the introduced *L. mollis* chromosomes in 17DM48. GISH revealed that 17DM48 had two alien chromosomes with green hybridization signals originated from the Ns subgenome of *L. mollis* (Figure 2a,b).

To determine the chromosome constitution of 17DM48, sequential FISH–GISH analysis was performed. Oligo-pSc 119.2 with green signals and Oligo-pTa535 with red signals, were able to distinguish the 42 wheat chromosomes simultaneously by combining these two oligonucleotide probes. Compared with the standard FISH karyotype of common wheat Chinese Spring (CS) [14], it was suggested that 17DM48 lacked a pair of wheat 2D chromosomes and presented two specific ones with the terminal strong binds distributed on both long and short arms by Oligo-pTa535 probe (Figure 2c). Sequential FISH-GISH analysis conducted on the same slide revealed that two extra specific chromosomes had strong signals of *L. mollis* (Figure 2d), thus corroborating that the lacked 2D chromosomes in 17DM48 were substituted by two Ns chromosomes of *L. mollis*.

### 2.3. Wheat 55K SNP Array Analysis

A wheat 55k SNP array were employed to further analyze the chromosomal composition of 17DM48. A total of 46,600, 48,033, and 28,361 polymorphic SNP loci were identified in 17DM48, 7182, and *P. huashanica*, respectively (Appendix A). The maximum, minimum, and mean percentages of SNP genotyping loci shared between 17DM48 and 7182 were 75.87% (on chromosome 1D), 12.37% (on 2D), and 53.62% overall. The maximum, minimum, and mean percentages of SNP genotyping loci shared between 17DM48 and *P. huashanica* were 29.02% (on 2D), 4.20% (on 4D), and 7.04% overall. Line 17DM48 had the lowest percentage of same SNP loci with its parent line 7182 but shared the highest one with *P. huashanica*. To make a visual comparison, the corresponding positions in chromosome 2D were marked, which had same genotype SNP loci in the same locations in line 17DM48, 7182, and *P. huashanica.* It showed that 17DM48 shared more of the same genotype SNP loci in the same locations as *P. huashanica* rather than line 7182 (Figure 3b). The result indicated that 17DM48 was a wheat–*L. mollis* 2Ns (2D) disomic substitution line.

### 2.4. Molecular Marker Analysis

PLUG and EST–STS markers distributed in seven homoeologous groups of wheat were used to verify the homoeologous relationship of alien chromosomes. The results showed that three PLUG markers (TNAC1139-TaqI/HaeIII, TNAC11204-TaqI/HaeIII and TNAC1210-TaqI/HaeIII) and three EST markers (BG607805-2AL/2AS/2BS, BQ169707-2AS/2BS/2DS, CD453246-2AS/2BS/2DS) located on the chromosomes of the second homoeologous group amplified expected bands in *L. mollis* and 17DM48 but not in common wheat line 7182 and the durum wheat line D4286 (Appendix A, Figure 4). It confirmed that alien chromosomes in 17DM48 belonged to the second homoeologous group. The polymorphic markers verified in this study can be used to trace chromosome 2Ns of *L. mollis* in wheat background.

### 2.5. Evaluation of Agronomic Traits

The agronomic traits of 17DM48 and its parents were investigated as shown in Table 1. In terms of resistance to stripe rust, line M842 were resistant to stripe rust with infection type 0, and its derived line 17DM48 displayed a high level of resistance with infection type 1 compared to its parent 7182 and the susceptible control Huixianhong (HXH) (Figure 5). The spike length of 17DM48 was significantly longer than all wheat parents. (Figure 6c). Similarly, the average number of florets and spikelets per spike of 17DM48 was extremely higher than those of its parents (Figure 6d,e).

### 2.6. Molecular Marker Development

A total of 6,643,408 reads were obtained from 17DM48, with the average GC percentage and Q30 percentage of 47.40% and 94.22% respectively. 466,330 SLAF numbers were predicted according to the enzyme digestion program. By using BWA and Local BLAST+ tools for analysis, 658 sequences were acquired with 0% similarity to the reference genome of CS (IWGSC-RefSeqv1.0) and 100% similarity to the simplified genome of *L. mollis* (unpublished data), which were considered to be specific sequences of *L. mollis*.

As expected, 80 primer pairs were designed based on these specific fragments and used for amplifying SLAF sequences from line 7182, D4286, *L. mollis* and 17DM48. Among them, 13 primers amplified specific bands between *L. mollis* and 17DM48 with a development success rate up to 16.25% (Figure 7, Table 2). These specific markers can be applied to quickly detect the genetic materials carrying 2Ns chromatin in wheat background, which was the first and successful case in developing efficient markers for detecting alien chromatin in *L. mollis*.

## 3. Discussion

Chromosome engineering plays an important role in developing novel germplasm and broadening the genetic base of wheat [15,16,17,18]. *L. mollis* is a beneficial resource for wheat improvement, with resistance genes against multiple fungal diseases and yield improvement potential [19]. The precondition of further utilization of derived lines from chromosome engineering is that alien chromosomes can be stably inherited to the progeny [20]. It is necessary to ensure the chromosome constitution as well as the genetic stability through cytogenetic observation of RTCs and PMCs during mitosis and meiosis, respectively. In this study, a novel wheat-*L. mollis* 2Ns (2D) substitution line was identified by GISH, FISH, 55K SNP array and molecular marker analysis. The cytology results indicated that 17DM48 had 42 chromosomes, with 21 bivalents in meiosis prophase I and without lagging chromosomes during the chromosome segregation in meiosis anaphase I, and then supported the genetic stability of 17DM48.

Due to the visualized features, GISH and FISH are usually the preferred methods for analyzing germplasm derived from chromosome engineering, and many derived lines, such as addition lines, substitution lines and translocation lines have been successfully identified using these molecular cytogenetic techniques [21,22,23,24,25]. In the study, GISH analysis indicated that a pair of Ns chromosomes from *L. mollis* were introduced into wheat background (Figure 2a,b). FISH revealed that 17DM48 possessed all chromosomes of common wheat expect two 2D chromosomes. But two chromosomes with unknown karyotype appeared in 17DM48, which were eventually demonstrated to be Ns chromosomes from *L. mollis* through a sequential FISH–GISH analysis on the same slide. The two alien chromosomes showed large clusters of red fluorescent signals at both ends of the chromosome arms and partially dispersive red signals in the middle (Figure 2c). Above-mentioned karyotype of 2Ns was firstly demonstrated to facilitate the subsequent material identification.

With the development of high-density loci arrays, SNP genotyping has gradually been widely used in gene localization and the determination of alien chromosomes introduced from wheat relatives [26,27]. A wheat-*Thinopyrum ponticum* 1Js (1D) disomic substitution line, and a wheat–*Psathyrostachys huashanica* Keng 5Ns (5D) substitution Line were analyzed respectively using wheat SNP arrays [12,28]. In the present study, wheat 55K SNP arrays were employed to determine the chromosome constitution of 17DM48. The results showed that in 17DM48, the same SNPs as its parent 7182 were deleted by 87.63% on chromosome 2D. On the contrary, compared with *P. huashanica*, the Ns subgenomic donor of *L. mollis*, 17DM48 had the highest proportion of the same SNP on chromosome 2D as the second homoeologous group chromosomes of *P. huashanica*, which indicated that a pair of 2Ns chromosomes originated from *L. mollis* substituted for wheat 2D chromosomes in 17DM48. Meanwhile, 3 EST-STS markers and 3 PLUG markers distributed in the second homoeologous group of wheat amplified target bands in 17DM48 and *L. mollis*, while no unique bands appeared in all other parents. These results confirmed that 17DM48 was a wheat–*L. mollis* 2NS (2D) substitution line, similarly supported by the results of FISH, GISH and SNP arrays.

The SLAF-seq technique, based on next-generation sequencing, has been applied to construct high-density genetic map, develop FISH probes, conduct polymorphism analysis and provide an efficient and convenient method for developing specific PCR-based markers for plants without a reference genome [29]. The increasing SLAF-based markers have been applied in chromosome engineering for rapidly tracing alien fragments. In the study, thirteen specific SLAF markers amplified specific bands in 17DM48 and *L. mollis* different from those in line 7182 and D4286, which contributes to tracing 2Ns chromosomes and further mapping of beneficial genes from wheat relatives.

Chromosome rearrangements during evolution widely existed in wheat relatives, including rye, barley, *Aegilops*, *Leymus* and other Triticeae species [30,31,32,33]. The evolutionary translocations broke down the collinearity between the homoeologous wheat and alien chromosomes, which may lead to the incomplete compensation [34]. In the study, SNP array indicated that chromosome 2Ns was homoeologous to wheat group 2 and possibly showed partial homoeology to three other groups (1, 5, 6). Previous studies reported that, rye chromosome arms 2RS, 3RL, 4RL, 5RL, 6RS, 6RL, 7RS and 7RL were involved in evolutionary translocations, which paired with wheat chromosomes from a different homoeologous group [32,35]. The reciprocal translocation between 4NsL and 5NsL had also been proved to occur in *Leymus* [36]. In this study, through karyotype comparison combined with the results of SNP array, chromosome arm 2NsS was found to show partial homoeology to 6DS, which was possibly caused by the reciprocal translocations between 2NsS and 6NsS in the process of evolution. As a likely consequence of the structural differences hence uneven gene dosages and incomplete compensation, it may explain the changes of spike morphology, low seed set and associated changes of kernel size of 17DM48. It was worth mentioning that 17DM48 possessed longer spike than all the parents, with an increased percentage of up to 71.93% compared to wheat line 7182. Further mining of corresponding genes contained in 17DM48 would contribute to the genetic improvement of wheat.

Stripe rust is a devastating wheat fungal disease around the world, which causes severe reduction in wheat production in a pandemic year. At present, more than 80 stripe rust resistance genes and related QTLs have been permanently designated in wheat and its relatives [37]. In previous study, several wheat-*L. mollis* derivatives from different homoeologous groups showed high resistance to stripe rust, involving 5Ns, 6Ns and 7Ns chromosomes [38,39]. Significantly, in wheat-*L.mollis* derived lines, both the double monosomic addition line (2*n* = 44 = 42T.a + L.m2 + L.m3) [40] and double substitution line (2*n* = 42 = 38T.a + 2L.m2 + 2L.m3) [41] were highly resistant to stripe rust, whereas the 3Ns(3D) substitution line lacked the stripe rust resistance instead with a leaf rust resistance [42]. It can be inferred that desirable resistant genes against stripe rust probably exist in 2Ns chromosomes of *L. mollis*. And more research is needed to isolate the potential resistance genes. Breeders prefer translocation lines with smaller alien fragments and less linkage drag, methodology is available to induce chromosome variation [24,43,44,45,46]. Finally, it is necessary to develop fine translocation lines involving small *L. mollis* fragments with desirable agronomic traits for further utilization in wheat improvement programs.

## 4. Materials and Methods

### 4.1. Plant Materials

The plant materials in the present study included bread wheat 7182 (2*n* = 6*x* = 42, AABBDD) and Huixianhong (HXH), *Leymus mollis* (2*n* = 4*x* = 28, NsNsXmXm), *Psathyrostachys huashanica* (2*n* = 2*x* =14, NsNs), octoploid *Tritileymus* line M842 (2*n* = 8*x* = 56, AABBDDNsNs), durum wheat line D4286 (2*n* = 4*x* = 28, AABB), and one wheat-*L. mollis* disomic substitution line 17DM48. The wheat line 7182 was employed as a control in the evaluation of agronomic characters, as well as in molecular marker analysis. The wheat cultivar HXH was used as a susceptible control in stripe rust evaluation. All the materials mentioned above were deposited in the College of Agronomy, Northwest A&F University (Yangling, China).

### 4.2. Cytological Observation

The seeds were germinated on moist filter paper in a Petri dish to obtain root tips at a suitable growth period. After nitrous oxide treatment, the root tips were fixed in 90% acetic acid for 10 min, which subsequently were stored in 70% ethanol at –20 °C for later use. By means of enzymatic hydrolysis with cellulase (R-10, Yakult Japan, Tokyo, Japan) and pectinase (Y-23, Yakult Japan, Tokyo, Japan) at 37 °C for 1 h, the root tips were made into suspension for sample making, using a production process described by Han et al. [47]. Young panicles at the appropriate stage were sampled, which later were processed with ethanol–chloroform–acetic acid mixture (6:3:1, *v*/*v*/*v*) for one week at 25–30 °C. Then, anthers were pinched out and crushed on a slide in 1% acetocarmine. Root tip chromosome number and pollen mother cells chromosome configuration were observed and photographed with an Olympus BX-43 microscope (Olympus Optical Co., Ltd., Tokyo, Japan) equipped with a Photometrics SenSys CCD camera DP80.

### 4.3. GISH, FISH and Sequential FISH–GISH

Before in situ hybridization, chromosomes were fixed for 60 s at an ultraviolet intensity of 125,000 mJ/cm2 by UV irradiation (Spectrolinker™ XL-1500, Long Island, NY, USA). The total genomic DNA of *L. mollis* and *P. huashanica* were used as probe for GISH analysis, which were labeled with Alexa Fluor 488-5-dUTP (Invitrogen, Carlsbad, CA, USA). The CTAB method [48] was performed to acquire highly purified DNA followed by a purification step employing a mixture of 25:24:1 phenol/chloroform/isoamyl alcohol. The GISH hybridization solution consisted of 0.3 L labelled probe DNA and 8.7 L GISH buffer (2 × SSC/1 × TE). After the hybridization droplets were added to the slides containing the cell split phase, the chromosomes were denatured along with the probe at 100 °C for 4 min, and finally renatured at 42 °C for more than 16 h [47]. The oligonucleotide probes Oligo-pSc119.2 (green) and Oligo-pTa535 (red) (Shanghai Invitrogen Biotechnology Co. Ltd., Shanghai, China) were used in combination for FISH and latter sequential FISH–GISH analysis [14]. After exposure treatment, GISH analysis was conducted on the same split phase as described earlier. Before microscopy, DAPI was employed to counterstain the chromosomes. Eventually, fluorescent signals were observed and photographed with an Olympus BX-53 microscope equipped with a Photometrics DP80 camera.

### 4.4. Wheat SNP Array Analysis

Genomic DNA of 17DM48 and its parents was hybridized to wheat 55K SNP genotyping arrays, and Illumina Bead Array technology was used for scanning in China Golden Marker Biotechnology Company (Beijing, China). The wheat 55K SNP array contained 49,078 SNPs distributing across 21 pairs of wheat chromosomes. The percentage between two lines containing the same genotype on each chromosome was calculated. Data analysis and graphing were performed to analyze polymorphic markers using Origin V9.1 (OriginLab Corporation, Northampton, MA, USA) and MapChart V2.32 (Wageningen University & Research, Wageningen, The Netherlands).

### 4.5. Molecular Markers Analysis

Expressed sequence tag–sequence-tagged site (EST–STS) markers (http://wheat.pw.usda.gov/SNP/new/pcr_primers.shtml, accessed on 23 December 2021) and PCR-based Landmark Unique Gene (PLUG) markers [49,50] distributed in homoeologous groups 1 to 7 of wheat chromosomes were synthesized by AuGCT DNA-SYN Biotechnology Co., Ltd. (Beijing, China). To confirm homoeologous group relationships of the introduced alien chromosomes in the wheat–*L. mollis* disomic substitution line 17DM48, all these molecular markers were employed for polymerase chain reaction (PCR) assays between 7182, D4286, *L. mollis* and 17DM48. The PCR procedures were carried out as described previously [51]. The PCR products were separated by 8% non-deformable polyacrylamide gel electrophoresis and visualized after silver staining [52]. The PCR products of PLUG markers were digested with HaeIII (37 °C) for 3 h or TaqI (65 °C) for 2 h to increase the polymorphism level. Subsequently, the digested products were separated by 2% agarose gel electrophoresis. The reagent for the PCR reaction was purchased from Takara Biomedical Technology (Beijing) Co., Ltd. (Beijing, China).

### 4.6. Molecular Marker Development

SLAF-seq of 17DM48 was performed by the Beijing Biomarker Technologies Corporation (Beijing, China). Qualified sample DNA was digested by the restriction enzyme HaeIII. The libraries were sequenced by the Illumina HiSeq platform after passing quality inspection. The data filtering steps are as follows: (1) remove reads with adapter sequences, (2) remove reads with more than 10% N content, (3) remove reads in which more than 50% of the bases had quality scores less than 10. Sequence quality and data volume were evaluated after filtering sequence readings [53].

Sequences specific to 2Ns of *L. mollis* were obtained by Burrows-Wheeler-Alignment V0.1.17 (BWA) [54] and Basic Local Alignment Search Tool V2.10.1 (ftp://ftp.ncbi.nlm.nih.gov/blast/executables/blast+/LATEST/, accessed on 13 August 2021). Sequences with 0% similarity to CS (IWGSC-RefSeq-v1.0) and 100% similarity to *L. mollis* (unpublished data) were selected for molecular markers development. The primers for these sequences were developed using Primer Premier V5.0 (PREMIER Biosoft, Palo Alto, CA, USA). After PCR, specific bands were detected on a 1% agarose gel.

### 4.7. Morphological Traits Evaluation

All materials were planted in the experimental field of Northwest A&F University, including 17DM48, parents M842 and D4286, as well as its previous ancestor 7182. At the physiological maturity stage, morphological traits of these lines were evaluated in 2020 and 2021. For each material, ten randomly selected plants were used for the evaluation of agronomic traits, including plant height, tillering, spike length, number of spikelets per spike, number of florets per spikelet, number of kernels per spike, thousand-kernel weight, and awnedness. In addition, each sample is measured for kernel traits through scanning 50 randomly selected kernels. Significant analysis between different materials were conducted using the SPSS Statistics 26 software program (IBM Corp., Armonk, NY, USA). Comparison and graphing of agronomic traits were performed using GraphPad Prism V8.0.1 (GraphPad Software, San Diego, CA, USA).

### 4.8. Disease Reaction Evaluation

7182, D4286, M842, 17DM48, HXH were evaluated for disease reaction to the fungal diseases stripe rust at adult stage. Mixed races of stripe rust fungus (CYR32, CYR33) were used for artificial inoculation. When stripe rust susceptible control variety HXH was fully infected, the response type was surveyed according to the previous standards [55]. The infection type (IT) standards of wheat stripe rust at adult stage were assessed on a 0–4 scale as follows: 0, immune; 0, nearly immune; 1, highly resistant; 2, moderately resistant; 3 and 4 as moderately susceptible and highly susceptible, respectively.

## 5. Conclusions

In this study, a novel wheat-*L. mollis* 2Ns (2D) disomic substitution line was identified using cytology methods, DNA markers, SNP array detection, GISH and FISH analysis. The developed line 17DM48 showed a high level of stripe resistance and longer spike. Based on SLAF-seq, thirteen specific markers were developed to identify and trace chromosome 2Ns of *L. mollis* in wheat background, thereby further promoting the process of fine mapping of stripe rust resistance genes and molecular marker-assisted breeding. In conclusion, this line has great potential in improving the stripe rust resistance of wheat. It could provide a novel germplasm to transfer stripe rust resistance genes for wheat breeding.

## Figures and Tables

**Figure 1 ijms-23-02676-f001:**
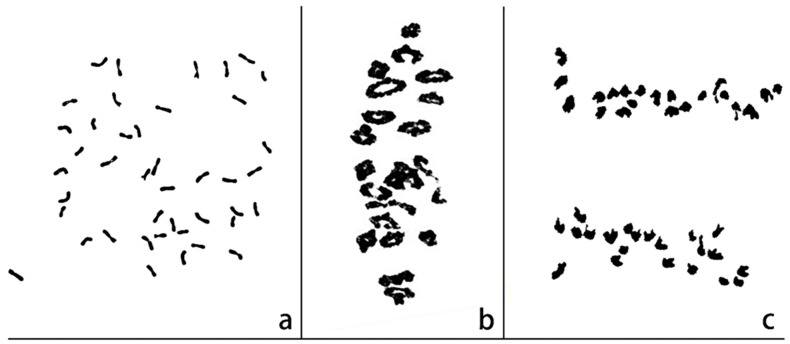
Cytogenetic analysis of 17DM48. (**a**) root tip cell at mitotic metaphase, 2*n* = 42. (**b**) chromosomal configuration of pollen mother cell at meiotic metaphase, 2*n* = 21 II. (**c**) chromosomal configuration of pollen mother cell at anaphase I, 2*n* = 21 + 21.

**Figure 2 ijms-23-02676-f002:**
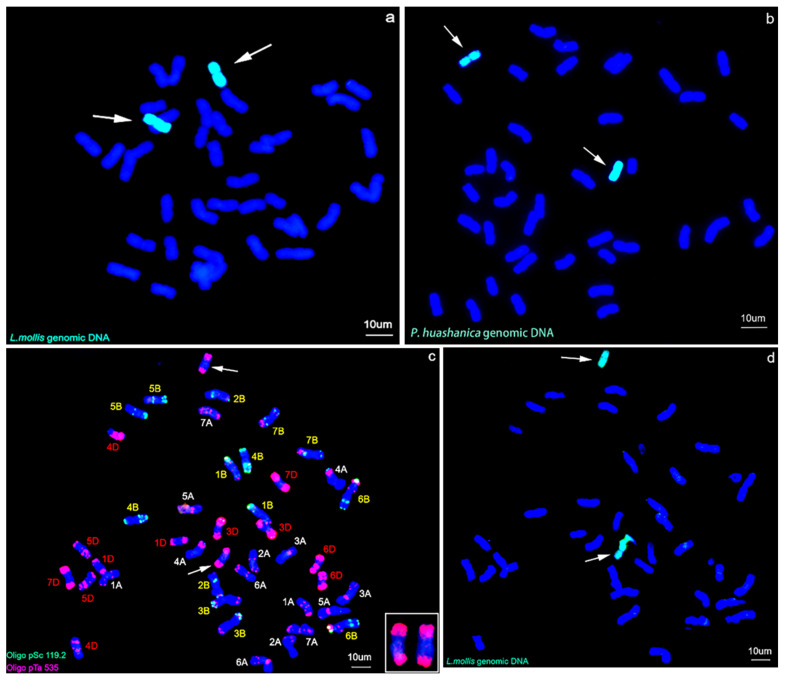
GISH and FISH-GISH analysis on 17DM48. The probes for GISH were *L. mollis* genomic DNA (green) and *P. huashanica* genomic DNA (green). (**a**) GISH detection of 17DM48 using *L. mollis* genomic DNA as probe (green). (**b**) GISH detection of 17DM48 using *P. huashanica* genomic DNA as probe (green). (**c**) FISH analysis of 17DM48. The probes were Oligo-pSc119.2 (green), and Oligo-pTa535 (red). (**d**) Sequential FISH-GISH analysis of 17DM48 by the probe of *L. mollis* genomic DNA. Chromosomes were counterstained using DAPI (blue). The arrows referred to alien chromosomes. Scale bar = 10 μm.

**Figure 3 ijms-23-02676-f003:**
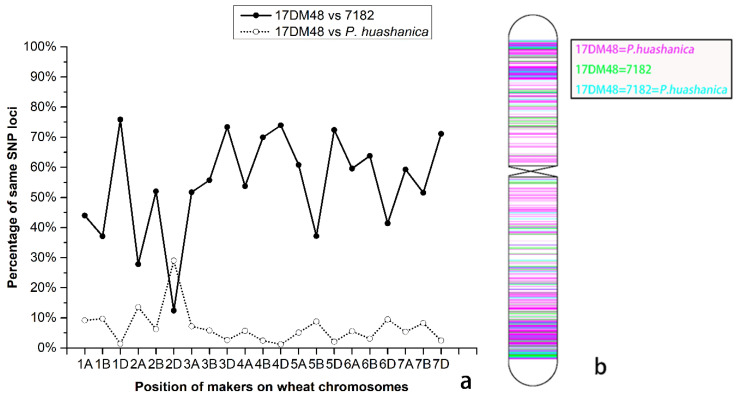
Wheat 55K SNP mapping analysis of 17DM48. (**a**) Obvious crossing point in terms of the position of the 2D chromosome. (**b**) Positions of the same SNP loci at the 2D chromosome in the genotype of 17DM48 with *P. huashanica* and line 7182.

**Figure 4 ijms-23-02676-f004:**
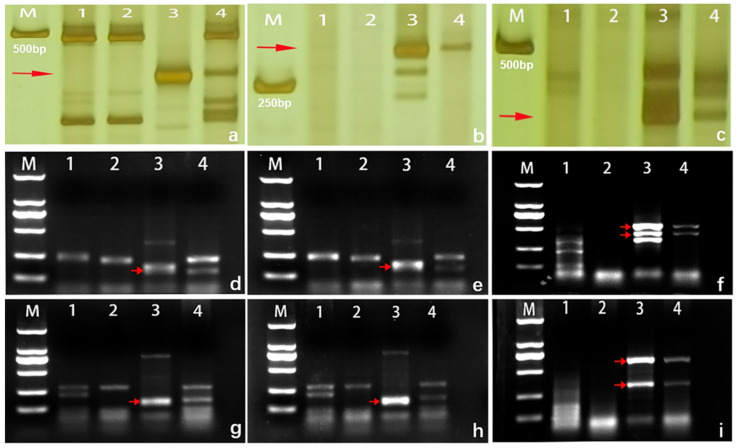
EST–STS and PLUG markers analysis of wheat-*Leymus mollis* disomic substitution line 17DM48. The red arrows indicate the *L. mollis* specific band. (M). DNA marker DL2000. (**1**) 7182. (**2**) D4286. (**3**) *L. mollis*. (**4**) 17DM48. (**a**) BG607805. (**b**) CD453246. (**c**) BQ169707. (**d**) TANC1204-TaqI. (**e**) TANC1210-TaqI. (**f**) TANC1139-TaqI. (**g**) TANC1210-HaeIII. (**h**) TANC1204-HaeIII. (**i**) TANC1139-HaeIII.

**Figure 5 ijms-23-02676-f005:**
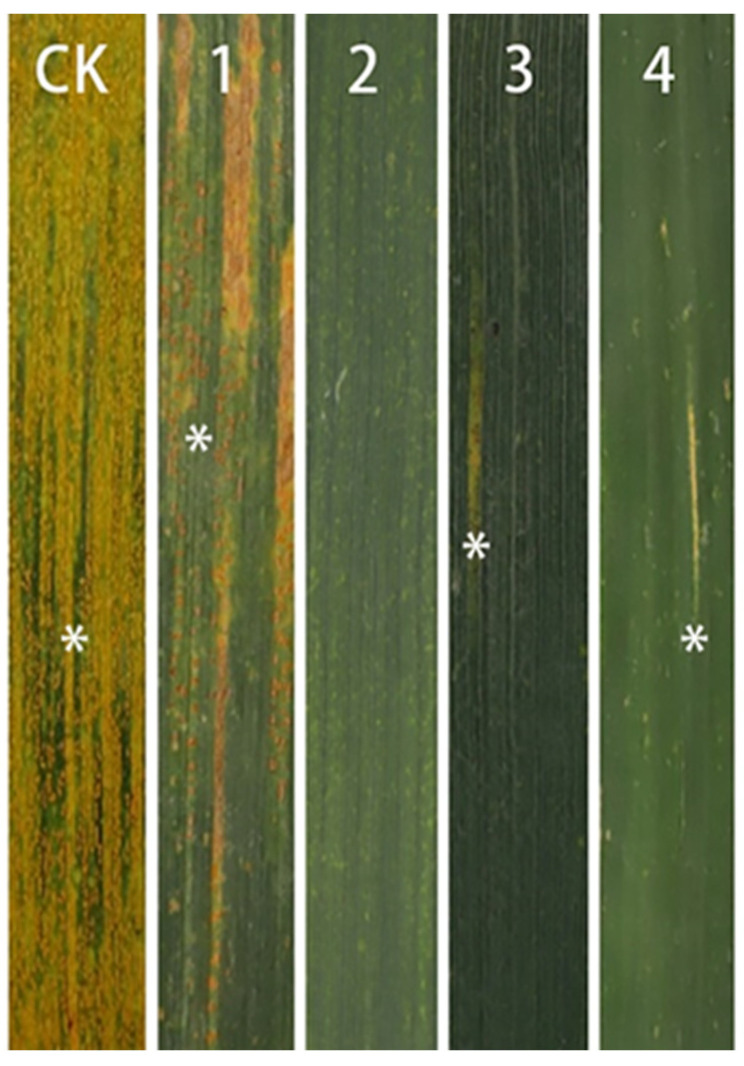
Stripe rust disease reaction of 17DM48 and its parents. (CK) *Triticum aestivum* Huixianhong. (**1**) *T. aestivum* cv. 7182. (**2**) octoploid *Tritileymus* M842. (**3**) *T. durum* D4286. (**4**) disomic substitution line 17DM48. Asterisks indicate the symptoms and types of stripe rust reaction.

**Figure 6 ijms-23-02676-f006:**
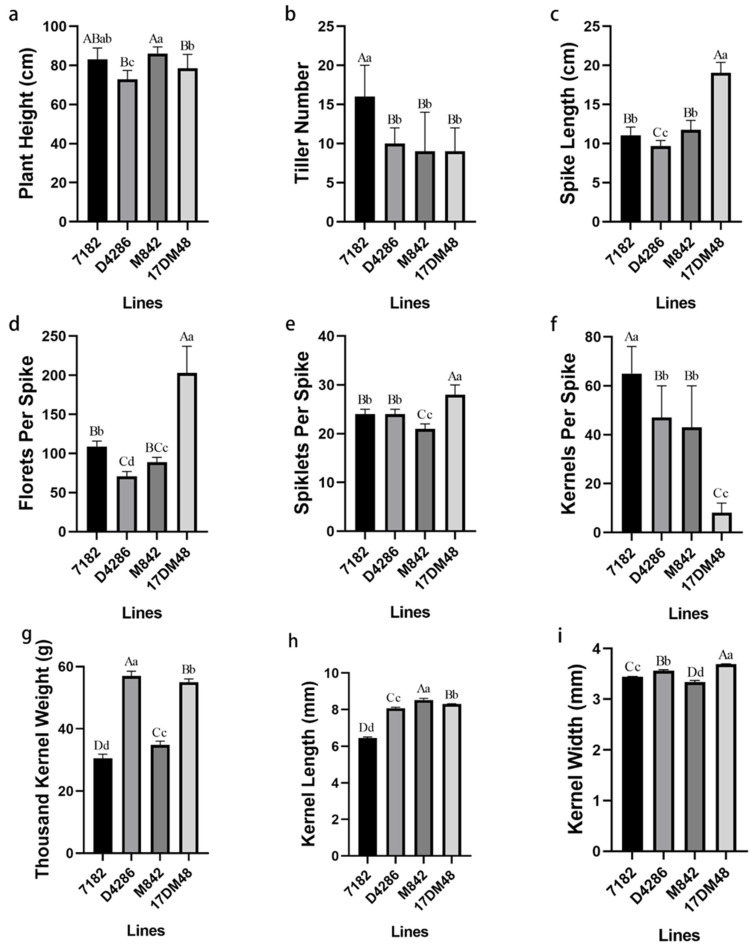
Analysis of agronomic traits of line 7182 and its parents. (**a**) plant height. (**b**) tiller number. (**c**) spike length. (**d**) florets per spike. (**e**) spikelets per spike. (**f**) kernels per spike. (**g**) thousand kernel weight. (**h**) kernel length. (**i**) kernel width. Capital and small letters indicate significant differences at *p* < 0.01 and *p* < 0.05, respectively.

**Figure 7 ijms-23-02676-f007:**
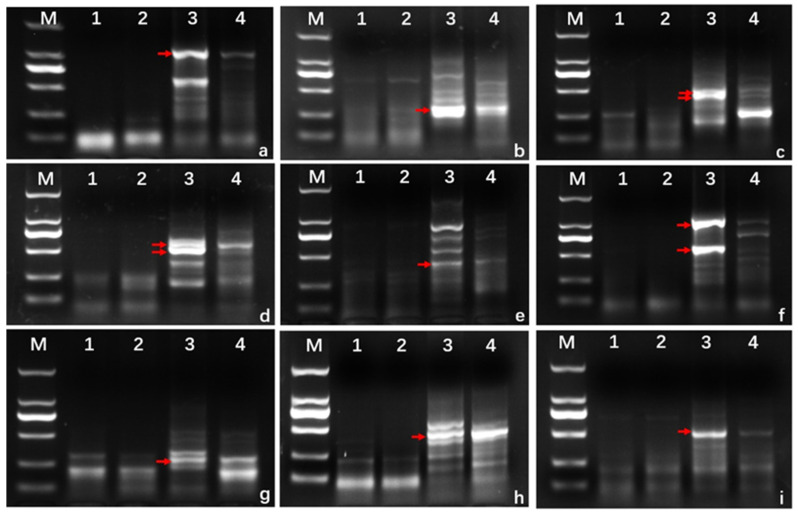
Molecular marker development and PCR amplification of wheat-*Leymus mollis* disomic substitution line 17DM48. The red arrows indicate the *L. mollis* specific bands. (M). DNA marker DL2000. (**1**) 7182. (**2**) D4286. (**3**) *L. mollis*. (**4**) 17DM48. (**a**) LM19474. (**b**) LM19428. (**c**) LM13006. (**d**) LM17228. (**e**) LM25058. (**f**) LM152390. (**g**) LM65677. (**h**) LM224473. (**i**) LM33865.

**Table 1 ijms-23-02676-t001:** Evaluation of agronomic traits of 17DM48 and its parents.

Material	Plant Height (cm)	Tiller Number	Spike Length (cm)	Florets Per Spike	Spikelets Per Spike	Kernels Per Spike	Thousand Kernel Weight (g)	Kernel Length (mm)	Kernel Width (mm)	Awn Type	Stripe Rust Reaction (IT)
M842	86.10 ± 3.32 Aa	9 ± 5 Bb	11.75 ± 1.21 Bb	89 ± 6 BCc	21 ± 1 Cc	43 ± 17 Bb	34.81 ± 1.27 Cc	8.53 ± 0.08 Aa	3.34 ± 0.03 Dd	short awn	0
D4286	72.88 ± 4.58 Bc	10 ± 2 Bb	9.69 ± 0.72 Cc	71 ± 6 Cd	24 ± 1 Bb	47 ± 13 Bb	57.01 ± 1.58 Aa	8.08 ± 0.05 Cc	3.56 ± 0.02 Bb	long awn	1
7182	83.15 ± 5.75 ABab	16 ± 4 Aa	11.08 ± 1.04 Bb	109 ± 7 Bb	24 ± 1 Bb	65 ± 11 Aa	30.53 ± 1.34 Dd	6.45 ± 0.06 Dd	3.44 ± 0.01 Cc	long awn	3
17DM48	78.55 ± 7.07 Bb	9 ± 3 Bb	19.05 ± 1.34 Aa	203 ± 34 Aa	28 ± 2 Aa	8 ± 4 Cc	54.95 ± 1.08 Bb	8.31 ± 0.01 Bb	3.69 ± 0.01 Aa	short awn	1
Huixianhong											4

Capital and small letters indicate significant differences at *p* < 0.01 and *p* < 0.05, respectively. IT, infection type.

**Table 2 ijms-23-02676-t002:** Specific molecular markers developed for 17DM48 based on SLAF-seq.

Marker	Tm (°C)	Primer (5′-3′)
LM19474	52	F: TCGTCTGGGTTTGCTTAT
		R: CACCGATTTCCAAGTTTC
LM19428	56	F: CGTCATCCTCCACCACCT
		R: ACGCAATCTGCTCAACCC
LM13006	56	F: TGCGGTTGCGTCTATTGG
		R: TGCTGGTGCATCATCATCG
LM17228	56	F: GCTCCTTTCTCGCTTGCT
		R: TGGACCGCTACGTTTGAC
LM25058	54	F: AGGAAGGGTCGGAAACTC
		R: AACACCACGGAATGAAGC
LM152390	52	F: TTTCTAGCCGCTAAAGGT
		R: TTTCCAAGCCTACTCCTG
LM65677	54	F: CAGAGCATAACCCAGGAG
		R: CCATAGGAACAAGCCAGA
LM224473	54	F: GGACGGTGAGCAAGAAGG
		R: CGTAATGCCCACGAAACA
LM33865	52	F: GCTAGTAAATCGGAGGAC
		R: TAGCCATAACACCAATCC
LM7529	52	F: AGGTTTCCAAATAAGGGAT
		R: CGGACCGTGAATACTCTG
LM12508	54	F: TCACGGCATACAACAAGG
		R: TATCCACCGACCACTCAA
LM23891	56	F: TGGGCAACCGATGCTCTA
		R: ACTGGCACGAATCCGTCT
LM51499	56	F: CAGCAGTGGCTTCTGTTCC
		R: TGTATGTGCGGGAGTGGA

## Data Availability

All data supporting the findings of this study are available within the paper and within its Appendix A published online. Further information may be obtained from the corresponding author, Wanquan Ji.

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
