# Peer review of "Identification and DNA Marker Development for a Wheat-Leymus mollis 2Ns (2D) Disomic Chromosome Substitution"

_ijms, 2022, doi:10.3390/ijms23052676_

Round 1

Reviewer 1 Report

In my judgement this is a standard run-of-the-mill paper on identification of a single alien chromosome substitution in wheat. Dozens, if not hundreds of such substitution had already been developed so in this sense, this article does not contribute anything new, apart from the first of this particular chromosome from this particular species. As a matter of fact, there is nothing on the development of the line; the paper focuses on chromosome identification.  Even here there is really nothing novel about the approach or methods used.

Technically, the paper is OK. As far as presentation, it is poor. The manuscript is very wordy; given that it presents very much a routine work, and of preliminary nature, it should be trimmed by about one half, with no information loss. English requires much attention, and it had better be from someone well versed in the system and terminology. The text is unusually adjective-rich. I strongly urge the authors to drop them all; do not push on a reader your fascination with your own work; let him/her decide alone what is so “novel”, “excellent”, “extremely significant”, “more significantly resistant” and the list goes on and on and on.  Just re-write the whole thing in standard English used in these types of articles, using standard phraseology and terminology. Drop these “obviously compared to its relatives” and similar. You do not “determine the homoeologous group”; you determine homoeology…. And, apparently, that homoeology is not nearly as great as you try to present it. The line has reduced fertility and that means incomplete compensation by 2Ns for 2D. So, my strong suspicion is that your 2Ns is only partially homoeologous to 2D and so very likely it is a translocated chromosome relative to wheat. Perhaps you ought to look at the signals from the array one more time to see if you can figure out what kind of translocation it really is. This incomplete homoeology easily explains spikelet branching; it is so common among wheat aneuploids….. Reduced seed set may also explain increased kernel size. If you do not believe it, emasculate a part of a spike of normal wheat, bag, and then compare the seed size to that from a fully fertile spike……

All of this above is yet another reason why all adjectives should be dropped. You may well be looking at artefacts rather than some fantastic contribution to wheat improvement. Perhaps only the rust resistance is of value and here you would have to find a way of extracting it from the chromosome to eliminate all deleterious effects of the whole chromosome substitution, with poor compensation.

In places I cannot understand the text, such as (l.140) “Further, a clear intersection could be seen between the two polylines in chromosome 2D…..”, and many others.

No, Leymus is not in the tertiary gene pool, at least not by the Western definition of gene pools. It does hybridize sexually with wheat; this is how you made your hybrid, did you not?

Be specific: do not use some cryptic line designations; these may mean much to you but mean nothing to a reader. If you say: triple substitution then list each one, and not a line designation. And so on.

As it is now typical, references are strongly biased toward Chinese. Looking at the list one would think that hardly anything has ever been done in the field before China joined it, rather recently. In actual fact, all principles and systems have been worked out by others, long long before. Yes, we all know that you are rewarded for citations but still…….

Below is my version of the abstract; the entire paper begs for the same treatment.

Leymus mollis (2n = 4x = 28, NsNsXmXm), a member of the secondary gene pool of wheat (Triticum aestivum L.), carries numerous loci which could potentially be used in wheat improvement. In this study, line 17DM48 was isolated from the offspring of a wheat x L. mollis hybrid. This line has 42 chromosomes forming 21 bivalents at meiotic metaphase I. Genomic in situ hybridization (GISH) demonstrated the presence of a pair chromosomes from the Ns genome of L. mollis. This pair substituted for wheat chromosomes 2D, as shown by FISH, by DNA marker analysis, and hybridization to wheat 55K SNP array. Therefore, line 17DM48 is a 2Ns(2D) disomic substitution line. It had reduced seed set, probably because of incomplete compensation by 2Ns, and associated increase in the 1000 kernel weight. It also shows longer spikes, spike branching, and a good level of stripe rust resistance. Using specific-locus amplified fragment sequencing (SLAF-seq), 13 DNA makers were developed to identify and trace chromosome 2Ns of L. mollis in wheat background. This line may provide a bridge for genetic improvement of wheat, especially stripe rust resistance, and may serve as a germplasm for genetic analysis of wheat spike morphogenesis.

Reviewer 2 Report

Minor remarks:

lines 26 - replace "thousand-kenel" by "thousand-kernel"

line 134 - delete "错误!未找到引用源"

Figure 2 caption - P. huashanica should be in italics line 158 - delete "错误!未找到引用源" lines 183-186 - the Table caption should be added. The table should be placed in the center of page line 352 HaeⅢ should be in italics  

Reviewer 3 Report

Dear Authors,

            I have a great honor and opportunity to review manuscript entitled: “Molecular Cytogenetics Identification and Specific Marker Development of Wheat-Leymus mollis 2Ns (2D) Disomic Substitution Line with Spike Branching and Stripe Rust Resistance” which is considered for publication in International Journal of Molecular Sciences (IJMS). I analyzed whole manuscript and it presents interesting approach to the spike Branching and stripe rust resistance but it also has some drawbacks which need improvement. The specific list of comments I present in a form of list presented below:

  1. General comments:

English language need improvements and also Edition of results. Authors need to check firstly the all publication rules associated with IJMS journal. Is beyond my comprehension  why authors send article which is not prepared according publication rules but in this case it happened. Authors must point by point check journal rules and adjust manuscript to this rules now article is sloppy. The most notorious problem is use of I think Chinese language in manuscript line 134, 158 and others. Moreover, any of 47 reference list position is not prepared according journal publication rules it must be corrected.

  1. Specific comments:
  2. Introduction section

This part of manuscript should introduce the scientific problem and what’s is the most important to present precise and scientifically formulated aim of the study or in case IJMS even aim and hypothesis of research. Currently the aim is not properly formulated I do not exactly also why authors make enumeration of objectives. Aim of the study must be formulated precisely not in a form of enymeration.

  1. Results section

Line 97-101. The observation process should be presented in a way that authors precisely name the used method and device.

Figure 1 is to low quality and too small for IJMS high standards

LINE 112- The citation should be avoided in results section. It is necessary to cited :  “Tang et al. 2014” must be cited as number which must be added in reference list

Figure 2 is divided into two separate pages why? The similar issue like in case Figure 1. This Figure is to low quality and too small for IJMS high standards and has so many bad pixels even on scale bars

Figure 3 I strongly recommend to change this graphical presentation it is not clear. I suggest to use the method of presentation used on Figure 3 in 10.3389/fpls.2021.689031. To show the look lie of chromosomes with they markings and markers on it

Table 1 and 2 should be moved to the supplementary data

Figure 4 must be enlarge

Figure 5 The symptoms must be marked on photos with use asterisk or arrow now the figure has not enough scientific information

Figure 6 must be enlarge currently size make the Figure low quality

Line 183 Table has no description this is an error. Moreover, all statistically significant values must be marked with bold and asterisk. It would be better to change this Table into chart with chart bars.

  1. Discussion

It is too short in context of amount of results this makes article not well balanced. This part must be extended

  1. Materials & Methods

All used solutions and software must have the name producer and country of producer. Moreover, In the case of software the versions of it must be added. If authors used any programs form Http websites the websites must be cited as reference with appropriate numbers as it is presented in Journal publication rules. I am sorry to say but authors present numerical data but any of paragraphs of this section did not describe the exact statistical analyses was performed.

Because, all of this issues I could not recommend publication of this paper current version.

Sincerely,

Reviewer 4 Report

The manuscript is well written. I have no additional comments. It can be accepted. 

Author Response

Thanks very much for taking your time to review this manuscript.

Round 2

Reviewer 1 Report

The authors clearly improved the presentation but they still refuse to address several key points which I raised in the previous review. I never doubted that the substitution was there and that it is stably inherited. This appears certain. But I believe there is a SERIOUS issue with compensation of the 2Ns for 2D. Just look at seed set: it is about 10% of that of the parents. So, your chromosome DOES NOT compensate fully for 2D. Actually, it compensates POORLY. Therefore, much of what you see agronomically may well be an artefact. If you refuse to test it before publishing then at least spell it out clearly. Instead, you keep beating on all that kernel size improvement! Of course these kernels are larger. It is a standard reaction to reduced seed set. I suggested a simple test, if you do not believe it. Now you also admit issues with lateness. I believe that this entire kernel size/weight should be dropped completely, or perhaps be summarized in a single sentence: "with seriously reduced seed set and much delayed maturity, whatever seed was set was larger and heavier than in the parents". It may well be the same story with branching. I see this in every single season when I grown unstable progenies from wide crosses of wheat and it always disappears once the lines are stabilized and compensation issues are resolved. Think of your 2Ns as translocated relative to wheat group 2 (or you may call it: "structurally different". Given the seed set of your line there is no doubt in my mind that it IS translocated. Look at literature data on, say, rye chromosomes 2, 3, 4, 6 and 7. Can you make stable substitution? Of course you can, and many exist. Are they fully fertile? No, they are NOT. Depending on how much of the rye chromosome is non-syntenic with wheat, the reduction can be minimal (such as  chromosome 2) or large (such as chromosomes 4 and 7). So, your chromosome shows homoeology to group 2, because, most likely, a good portion of it actually is genetically equivalent to wheat group 2. But other portions are not.  They are equivalent to some other homoeologous groups of wheat. Therefore, your line is deficient for some sets of genes and has extra copies of some other sets of genes, with all consequences that come with it. 

I suggested looking again at your array data. If you analyze it carefully you MAY be able to detect segments of homoeology along that 2Ns, based on the strength of the hybridization signal. It DOES work.....

In the process of toning things down I suggest this title: "Identification and DNA Marker Development for a Wheat-Leymus mollis 2Ns (2D) Disomic Chromosome Substitution"

English: I still see numerous problems; the most jarring its that "posterity". Where did this come from? How about "progeny"?

My comment about citations stands unaltered. Actually, I do not see any changes there, and I still consider it unfair. 

Reviewer 3 Report

Dear Authors,

I thank you for your responses  recomend publication.

Sincerely

Author Response

(The authors gave the same response as above.)

Round 3

Reviewer 1 Report

After reading your response to my comments I assumed we have got there.  But only until I read the manuscript. Yes, I understand that we all get very attached to our words but at times there is time to cut/delete and be done.  If you looked at your chip data and now you can see evidence of structural differences between your 2NS and 2D, with partial homoeology to three other homoeologous groups of wheat, and you can see why fertility is low, and when you can safely assume that much, if not all, branching is due to structural differences, why do you still insist on working the kernel size (Fig 6) and why do you keep working that branching? Why not start the Results from the chip data, by showing that the chromosome is partially homoeologous to wheat gr. 2, and three other groups, and discussing structural karyotype rearrangements in evolution (there are tons of examples now, not only from rye; just look up Jan Dvorak and Ming-Cheng Luo) and explain how these structural differences may affect compensation and morphology. You can dismiss with a single sentence both branching and kernel size (as a likely consequence of structural differences hence uneven gene dosages), drop the paragraph 238-245, and Figure 6, and just stick to valid points. The only agronomically valuable effect here is that rust resistance. Getting to it may be another matter, as you are dealing with a structurally rearranged chromosome that may refuse to pair and recombine with wheat. And there are examples of that too, just look at rye vs. wheat. Look up Tomas Naranjo on how individual rye arms pair with wheat, and put that together with Katrien Devos on structural differences between rye and wheat. Actually, if I were you I would dismiss the entire story of agronomical value with a statement that agronomically the line is of little value, because of structural chromosome differences and incomplete compensation, with the exception of rust resistance. So, this sentence for the Abstract "This line exhibited spike branching, stripe rust resistance and other excellent agronomic traits like longer sipke, numerous spikelets and high thousand-kenel weight." is simply untrue, especially that "excellent" part. 

I do think that you have a neat little paper here, and it is worth publication. But drop stuff that only muddles things up. Instead, how about a nice figure of the structure of 2Ns? You should be able to tell now which segment of it corresponds to which wheat group, and in what location! Wouldn't that be a neat illustration? Perhaps with some digging you can start guessing where your rust resistance locus is (or, rather, may be).

Just make it into a neat tight paper. You are close!
